# Tissue-Based Diagnostic Biomarkers of Aggressive Variant Prostate Cancer: A Narrative Review

**DOI:** 10.3390/cancers16040805

**Published:** 2024-02-16

**Authors:** Olga Kouroukli, Vasiliki Bravou, Konstantinos Giannitsas, Vasiliki Tzelepi

**Affiliations:** 1Department of Pathology, Evaggelismos General Hospital, 10676 Athens, Greece; 2Department of Anatomy-Histology-Embryology, School of Medicine, University of Patras, 26504 Patras, Greece; vibra@upatras.gr; 3Department of Urology, School of Medicine, University of Patras, 26504 Patras, Greece; giannitsasko@upatras.gr; 4Department of Pathology, School of Medicine, University of Patras, 26504 Patras, Greece

**Keywords:** AVPC, biomarkers, NEPC, transdifferentiation, ADT, tumor suppressors, DDR, epigenetic regulation

## Abstract

**Simple Summary:**

Metastatic prostate cancer is traditionally treated with androgen deprivation therapy. The introduction of second-generation antiandrogens into clinical practice elicits prolonged responses but also gives rise to mechanisms of resistance that do not rely on androgens. The emergent phenotype of androgen-indifferent prostate cancer is associated with an aggressive, atypical clinical course. The term “aggressive variant prostate cancer” (AVPC) has been coined in order to separate this phenotype from hormone-responsive tumors. Unfortunately, morphology alone cannot reliably predict virulent behavior. The development of prognostic and predictive biomarkers is therefore crucial. In line with this, research has been focusing on unraveling the biological identity of AVPC. Drawing from the current knowledge about AVPC molecular pathogenesis and evolution, we attempt to identify candidate tissue-based AVPC biomarkers.

**Abstract:**

Prostate cancer (PC) is a common malignancy among elderly men, characterized by great heterogeneity in its clinical course, ranging from an indolent to a highly aggressive disease. The aggressive variant of prostate cancer (AVPC) clinically shows an atypical pattern of disease progression, similar to that of small cell PC (SCPC), and also shares the chemo-responsiveness of SCPC. The term AVPC does not describe a specific histologic subtype of PC but rather the group of tumors that, irrespective of morphology, show an aggressive clinical course, dictated by androgen receptor (AR) indifference. AR indifference represents an adaptive response to androgen deprivation therapy (ADT), driven by epithelial plasticity, an inherent ability of tumor cells to adapt to their environment by changing their phenotypic characteristics in a bi-directional way. The molecular profile of AVPC entails combined alterations in the tumor suppressor genes retinoblastoma protein 1 (RB1), tumor protein 53 (TP53), and phosphatase and tensin homolog (PTEN). The understanding of the biologic heterogeneity of castration-resistant PC (CRPC) and the need to identify the subset of patients that would potentially benefit from specific therapies necessitate the development of prognostic and predictive biomarkers. This review aims to discuss the possible pathophysiologic mechanisms of AVPC development and the potential use of emerging tissue-based biomarkers in clinical practice.

## 1. Introduction

Prostate cancer (PC) is the second most common malignancy in men worldwide. Well-established risk factors include advanced age, race, with higher incidences in African-American men, family history, and inherited genetic factors [1]. Dietary factors [2] and physical activity [3] influence PC development and progression, with several studies linking obesity with a higher risk of PC initiation or a tendency towards more aggressive disease [4,5,6,7]. The effects of smoking, alcohol consumption, chronic inflammation, STDs, and vasectomy have also been studied [1]. Environmental factors are also relevant, as evidenced, for example, by the association of low tissue concentrations of zinc and iron with aggressive PC [8].

At the initial stages of oncogenesis, PC is an androgen receptor (AR)-driven disease [9], treated systematically with androgen ablation. Eventually, a castrate-resistant state is reached, where the disease progresses despite castrate levels of androgen [10]. In most cases, resistance continues to involve AR-dependent mechanisms, temporarily counteracted by new androgen-targeting drugs, enzalutamide (a potent AR inhibitor) and abiraterone (an androgen biosynthesis inhibitor) [11]. However, a subset of castration-resistant PCs (CRPCs) evolves into AR-indifferent disease with inherent resistance to androgen deprivation therapy (ADT) [12]. The aggressive variant prostate cancer (AVPC) is the current umbrella term for clinically virulent AR-independent tumors, encompassing neuroendocrine PC (NEPC) and double negative PC (DNPC) [13]. NEPC, in the vast majority of cases, emerges after treatment (t-NEPC) of a prostate adenocarcinoma, the prototype of PC (from here on referred to as PCA), rather than de novo [14]. It is estimated that at least 20–30% of metastatic CRPCs (mCRPCs) progress to NEPC. NEPC represents the extreme end of AR independence with lineage switching, the acquisition of neuroendocrine (NE) markers, and the activation of alternative survival mechanisms [15]. DNPC, on the other hand, are tumors lacking both AR and NE marker expression that possibly correspond to a transitional state from AR-dependent disease to NEPC [16]. It appears that biologic heterogeneity, even within the AVPC group of tumors [17] is a real challenge in the attempt to accurately define and recognize these tumors. This is further exacerbated by the paucity of diagnostically valuable biopsy samples in the context of advanced, metastatic disease [18]. However, the need to identify AVPC patients is becoming increasingly relevant considering (1) the expected rise in AVPC incidence by the wider implementation of new, efficient antiandrogens in clinical practice [19] and (2) evidence of benefit from chemotherapy combinations for this subset of patients [20,21]. In this review, we intend to describe biomarkers emerging from current knowledge regarding AVPC molecular pathogenesis that could identify high-risk PC patients and guide therapeutic decisions. 

## 2. AVPC Definition

AVPC was conceptualized after the clinical observation that a subset of CRPCs followed a non-conventional clinical course reminiscent of that of small cell PC (SCPC), with predominantly visceral rather than bony metastases, lytic rather than osteoblastic bone disease, and relatively low prostate-specific antigen (PSA) levels. The hypothesis that these tumors, in addition to clinical features, also show similar biology and chemo-responsiveness to SCPC was tested with a phase II clinic trial of first-line carboplatin-docetaxel and salvage etoposide-cisplatin. The eligibility criteria for patients’ selection were the following: (1) SCPC histology, (2) exclusively visceral metastatic spread, (3) predominantly lytic bone metastases, (4) bulky (≥5 cm) lymph node mass or bulky (≥5 cm) mass in prostate/pelvis with Gleason score (GS) ≥8, (5) low PSA (≤10 ng/mL) at first presentation (before ADT) or at symptomatic progression during ADT despite high volume (≥20) bone metastases, (6) positive immunohistochemistry (IHC) for NE markers (chromogranin A or synaptophysin) or abnormally elevated serum NE markers (chromogranin A or gastrin-releasing peptide) at initial presentation or progression together with non-otherwise explained serum LDH and/or CEA ≥ 2× upper normal value and/or malignant hypercalcemia, (7) short interval period (≤6 months) between ADT initiation and AR-independent progression. Except for patients with a histologic diagnosis of SCPC, all others were required to have undergone ADT, have progressed during treatment, or have an unsatisfactory response (Table 1). This set of aggressive features was used to define clinicopathologic AVPC (AVPC-c) [20]. The term “anaplastic PC” that was previously used to describe patients with similar virulent features was abandoned on the grounds that the term “anaplasia” from a pathologist’s viewpoint implies cellular pleiomorphism [22]. A high percentage of patients fulfilling at least one of the AVPC-c criteria responded, although briefly, to platinum-based combination therapy. Overall survival (OS) was significantly shortened in relation to the number of fulfilled criteria [20]. Subsequent clinical trials reinforced the notion of platinum efficiency in AVPC. The addition of carboplatin to cabazitaxel was selectively beneficial for AVPC patients compared to non-AVPC patients [21]. In another study, platinum-based therapy could counterbalance the poorer prognosis of AVPC by achieving similar progression-free survival (PFS) and OS scores with conventional CRPC [23]. 

The molecular signature of AVPC was investigated in AVPC tissue samples and patient-derived xenografts (PDX). Molecular AVPC (AVPC-m) is defined by the combined loss of tumor suppressors retinoblastoma protein 1 (RB1), tumor protein 53 (TP53), and/or phosphatase and tensin homolog (PTEN) (≥2/3), as this molecular profile was more frequent in AVPC-c (48.3%) compared to unselected CRPC patients (26%). Morphology did not predict the underlying molecular changes [24]. 

AVPC spans a spectrum of histological appearances, from poorly differentiated adenocarcinoma to mixed NEPC-PCA to pure NEPC, either SCPC or large cell neuroendocrine prostate carcinoma (LCPC) [22] (Figure 1). Unusual features, such as aberrant squamous differentiation, have also been described [25]. AVPC with PCA histology usually shows a typical GS 5 architectural configuration with nested or solid growth without lumina formation and with prominent nucleoli [26]. SCPC is histologically distinctive with a high nuclear-to-cytoplasmic ratio, nuclear molding, indistinct cell borders, a lack of nucleoli, prominent apoptotic and mitotic activity, and necrosis. Positivity for at least one NE marker [synaptophysin, chromogranin A, CD56, or the newer NE marker insulinoma-associated protein 1 (INSM1)] is confirmatory (Figure 1). LCPC is rare and is characterized by large nests of cells with peripheral palisading, prominent nucleoli, and NE marker expression [22]. Histologic diagnosis of NEPC in the context of prior therapy is by definition conclusive of AVPC. The challenge lies in predicting which cases of morphologically conventional PC would have aggressive behavior or benefit from specific treatments. 

## 3. NE and AR-Signaling Markers 

Taking into consideration that a subset of AVPC show positivity for NE markers and that elevated serum chromogranin A has been previously reported to have an unfavorable prognostic role in CRPC patients [27,28] it seems reasonable to consider NE markers as possible biomarkers for AVPC detection. NE cells are normally scattered in small numbers in the basal cell layer of prostate acini [22]. Focal expression of NE markers is seen in 10–100% of usual PCAs [22] (Table 2) and although it has been associated with poorly differentiated tumors, it does not represent an independent prognostic factor [29,30]. NE cells in PCA do not proliferate, but they secrete peptides that may boost the survival of adjacent cancer cells [31]. The expansion of NE cells that occurs after prolonged exposure to ADT has been hypothesized to confer resistance to therapy due to the delivery of alternative, AR-independent survival signals [32]. Serum or tissue chromogranin A or synaptophysin expression did not show, however, a strong correlation with progression-free survival (PFS) or OS in AVPC-c patients. Expression of NE markers may represent an epiphenomenon rather than a driver of aggressive disease [20]. It has also been repeatedly shown that serum NE markers alone cannot predict response to chemotherapy [33,34,35,36].

Downregulation of AR and AR-regulated genes, such as PSA, transmembrane serin protease 2 (TMRSS2) and NK3 homeobox 1 (NKX3.1), along with loss of AR expression, in SCPC is consistent with AR-independent growth [37,38,39] (Figure 1). Reduced AR staining (<10%) is also observed in 36% of AVPC [24]. On the contrary, frequent, intense AR expression has also been reported in therapy-related SCPC (t-SCPC). Rates of intense AR staining did not differ significantly between t-SCPC and castration-resistant PCA (75% and 87%, respectively), despite lower AR transcriptional activity in t-SCPC. Epigenetic regulation may explain this discordance [40]. “Atypical” SCPCs with retained AR signaling activity have also been described by other groups [41]. In AR-positive AVPC cases, downstream molecules of AR signaling were co-expressed [24]. In addition, AR staining does not necessarily predict response to ADT [42]. Levels of AR expression and AR transcriptional scores between t-NEPC and castration-resistant PCA patients significantly overlap [43]. Apparently, there is a spectrum of AR expression and activity in CRPC that cannot reliably distinguish AVPC patients (Table 2). AR-independent resistance mechanisms in AVPC are implied by the paucity of AR splice variants [24,43] and the absence of AR activating mutations and amplification in t-NEPC compared to castrate resistance PCA [43].

Transcriptome analysis of mCRPC samples revealed an inverse relationship between AR and NE signatures. Low AR signaling and high NE transcript levels were mostly associated with tumors with a NE histology. Notably, some cases of low AR and high NE scores, as well as some of intermediate AR and NE scores, corresponded to adenocarcinomas with distinctive atypical nuclei, possibly in transition to NEPC [44]. 

DNPC are tumors with an AR−/NE− phenotype, with an increase in their prevalence from 5 to 20% of PC after the introduction of the new antiandrogens into clinical practice. Experimental models reveal that these tumors develop from clones that have resisted strong AR inhibition and rely on alternative survival pathways, namely fibroblast growth factor receptor (FGFR)—mitogen-activated protein kinase (MAPK) signaling [16]. Labrecque et al. classified mCRPC into five phenotypes based on IHC and transcriptional expression of AR and NE markers: (1) AR+ (high)/NE− (ARPC), (2) AR+ (low)/NE− (ARLPC), (3) AR+/NE+ (amphicrine), (4) AR−/NE− (DNPC), (5) AR−/NE+ (SCPC) (Table 2). Each phenotype is enriched for different biological properties, with DNPC and SCPC sharing a high metastatic potential. These phenotypes are dynamically related. Most importantly, DNPC has an intrinsic ability to transform into SCPC and may embody a transitional state to NEPC [45] (Figure 2). Squamous differentiation, previously reported in AVPC [25], was seen in a subset of DNPC. The above evidence discloses the complementary role of AR signaling and NE profile analysis in understanding the heterogeneous biology of CRPC.

## 4. Tumor Suppressors RB1, TP53, and PTEN

Loss of at least two of the tumor suppressors RB1, TP53, and PTEN characterizes half of the AVPC-c cases [24] and is associated with clinically significant benefit from chemotherapy. Indeed, combined defects in at least two of the three genes, termed unfavorable genomics [21] are included in the NCCN criteria for adding cabazitaxel to carboplatin [46].

Immunohistochemical detection of defects in two of these three molecules predicted improved responses to chemotherapy [21] and accurately categorized cases as AVPC-m (Table 2). A cut-off value of 10% was used to define gene loss by IHC (≤10% for RB and PTEN and ≥10% for p53, the latter due to nuclear accumulation of the mutated protein), and intensity of staining was considered to achieve optimal correlations with loss-of-function transcriptional scores. Intense (2+ 3+) staining was used to define abnormal RB1 and p53 expression, while any staining (1+–3+) defined abnormal PTEN expression [47]. In contrast to TP53 missense mutations that lead to p53 nuclear accumulation, and, thus, intense p53 staining by IHC (Figure 1), nonsense, frameshift, and indel alterations were not easily detected by IHC due to the low basal p53 expression in normal prostate and p53-wild-type PC [48]. Next-generation sequencing (NGS) remained a valuable tool for identifying TP53 alterations [47], although IHC proved to be just as, if not more, accurate as NGS. 

A large number of clinical and pre-clinical data suggest that the use of RB1, p53, and PTEN as biomarkers of AVPC in clinical practice is not only accurate and feasible but also biologically relevant. RB1 deletions and microdeletions represent a well-established genetic event in SCPC [38,43,49] resulting in an almost universal absence of protein expression by IHC [38,49]. RB1 allelic loss is extremely rare in primary PC but relatively common in mCRPC [49,50]. The retained RB1 expression in these mCRPC cases, along with its loss in NEPC, suggests a driver role of RB1 protein loss in NEPC, further highlighted by the frequently observed concurrent loss of RB1 expression in the adenocarcinoma component in mixed PCA-NEPC [38,49]. Evidence supports a common origin of the two components in mixed tumors [51,52] and NEPC emergence through transdifferentiation of PCA under the pressure of ADT [43,53]. Loss of RB1 protein seems to predict transition to NEPC as it precedes the morphologic shift of PCA to NEPC [38] (Table 2). Furthermore, the presence of RB1 alterations has been identified as a poor prognostic factor in mCRPC [44] (Table 2). Functionally, RB1 loss relieves inhibition of E2F target genes and not only enables cell cycle progression but also deregulates AR expression [50,54].

RB1 alterations tend to be mutually exclusive with AR alterations [44] but frequently coincide with TP53 mutations [24,43,44]. Concurrent RB1 and TP53 alterations are seen in 74% of mCRPC with a NE phenotype, in contrast to 5% of primary PC and 39% of mCRPC-Adeno [55]. Combined RB1 and TP53 inactivation is prevalent in poorly differentiated NE tumors [56] and has been reported to induce small-cell lung cancer [57]. In experimental models of PC, while knockout of either RB1 or TP53 failed to give rise to invasive carcinomas [58,59,60,61], simultaneous inactivation of both molecules resulted in poorly differentiated PCs with co-expression of luminal and NE markers, de novo ADT resistance, and visceral metastatic spread [61,62], features reminiscent of AVPC. In transgenic mouse models of prostate adenocarcinoma (TRAMP), the SV40 large T antigen, which binds and inactivates RB1 and p53, induced NE-like, metastatic tumors [56,63]. While other oncogenic events can initiate poorly differentiated PC from prostate basal cells, additional defects in RB1 and/or TP53 are required for SCPC emergence [64]. RB1 and p53 act as guardians of lineage commitment [65,66,67,68,69], so their inactivation enables lineage plasticity and reprogramming of PC cells. In the LNCaP/AR and CWR22Pc-ER models, concurrent RB1 and p53 loss conferred AR-independent resistance to enzalutamide with expression of basal and neuroendocrine markers and suppression of luminal markers. This transition was mediated by the transcription factor (TF) SRY-Box transcription factor 2 (SOX2) [55], previously found to induce pluripotency [70], and was rapidly reversible after the return of RB1 and p53 to baseline levels. This suggests a bidirectional, universal passage of tumor cells from a dedifferentiated, flexible state, marked by loss of luminal markers [55] and is consistent with the theory of DNPC being a transitional state (Figure 2). RB1 and p53 loss-of-function alterations may determine transcriptionally and epigenetically the conversion to NEPC. This dual inactivation has a synergistic effect on increasing chromatin accessibility in genomic regions crucial for neuronal differentiation and, accordingly, decreasing accessibility to genes related to PCA and epithelial differentiation [64]. An additional mechanism of synergy may be the unhindered cell proliferation caused by the “double hit” at cell cycle checkpoints. RB1 regulates the transition from the G1 to the S phase, so inactivation of RB1 leads to uncontrollable cell cycle progression that would be otherwise preventable by a functional p53 [61].

TP53 alterations are frequent in AVPC and SCLC [24,49,71] and mostly include deleterious mutations that extend the protein’s half-life, resulting in nuclear accumulation, detected by IHC [47,49]. In most studies, TP53 alterations are encountered in <10% of primary PC [53,72] but are significantly enriched in advanced disease [53,72,73,74,75,76]. TP53 defects are an independent poor prognostic factor among CRPC patients [48,77,78] (Table 2) and are associated with rapid acquisition of resistance to abiraterone or enzalutamide [62,77,78]. P53 abnormal staining, when found in the initial prostate biopsy or prostatectomy specimen, could be used as an early predictor of aggressive progression and resistance to enzalutamide or abiraterone [48,76,77]. TP53 knockout, however, does not suffice to induce resistance to enzalutamide [55]. Certainly, TP53 could not be independently used as a specific biomarker for AVPC [49] (Table 2).

In contrast to TP53 and RB1 alterations, PTEN deletion is a recurrent genomic alteration of primary PC (Table 2), documented decades ago [79,80]. PTEN loss is capable of PC initiation [81,82,83] and is frequently hemizygous [84,85], consistent with the identification of PTEN as a happloinsufficient gene [86]. Interestingly, bi-allelic PTEN deletion induces a p53-mediated program of cellular senescence that restrains tumorigenesis [60,87,88]. Homozygous PTEN deletion is more common in CRPC [74,89] where it is often coupled with TP53 loss [53], resulting in dysregulated senescent response and lethal tumor progression [60]. The detrimental effect of combined PTEN and TP53 loss was validated with the PBCre4^+^:Pten^fl/fl^:TP53^fl/fl^ model. The aggressive biology of tumors in this model was attributed not only to decreased senescence but also to the induction and increased plasticity of prostate stem cells [90]. Loss of PTEN and TP53 leads to histologically diverse tumors [53,90] that are phenotypically and molecularly related to human CRPC-NE and have intrinsic resistance to abiraterone [53].

In PTEN-deficient tumors, RB1 deletion facilitates visceral metastatic spread and lineage plasticity, as evidenced by the heterogeneous cellular composition of the emerging tumors with variable expression of cytokeratin, AR, and synaptophysin. The addition of TP53 loss resulted in rapidly metastasizing, lethal tumors with de novo ADT-resistance. Both PBCre4^+^:Pten^fl/fl^:Rb1^fl/fl^ and PBCre4^+^:Pten^fl/fl^:Rb1^fl/fl^:Trp53 ^fl/fl^ models recapitulate the molecular signature of human NEPC [62].

It becomes therefore apparent that combined defects in tumor suppressors potentiate lineage plasticity, the acquisition of ADT resistance, and other clinical and molecular features of AVPC. The combination of tumor suppressor gene alterations as biomarkers achieves higher specificity than either of these biomarkers alone (Table 3). The value of tumor suppressor as prognostic biomarkers in PC was highlighted by a recent study declaring the superiority of tumor suppressors alterations detection compared to traditional prognostic parameters in early prediction of aggressive disease course [91].

## 5. Oncogenes MYCN and AURKA

N-myc proto-oncogene protein (N-Myc), encoded by MYCN Proto-Oncogene, BHLH Transcription Factor (MYCN), is a transcription factor essential for brain development that is not expressed in normal prostate epithelium. MYCN is amplified in neural, hematologic, and other tumors, including 40% of NEPC and a small percentage of CRPP-Adeno [37,94,95,96]. MYCN amplification has been linked with poor prognosis in both NEPC and CRPC-Adeno [95] and the timing of its occurrence coincides with metastatic spread [43]. N-Myc-driven AVPCs have been reported as poor responders to docetaxel [17] (Table 2), the standard of care for mCRPC [97]. N-Myc downregulates AR expression and directly suppresses AR-targeted genes, providing a mechanism for abrogating AR dependence [37,95,98]. A role for N-Myc in lineage determination through epigenetic reprogramming has also been suggested. Following castration, the N-Myc signature shifts and is enriched with neural lineage and stem cell programs. N-Myc also biases bivalent H3K27me3 and H3K4me3 histone marks toward neural lineage gene activation [98] and directly upregulates NE markers [37]. The N-Myc signature is, thus, consistent with the NEPC molecular program [95] and could be used to predict CRPC patients at risk of developing NEPC [98] (Table 2). 

Supporting its role in lineage plasticity, N-Myc overexpression, combined with activated AKT serine/threonine kinase 1 (AKT) or PTEN loss, induces invasive tumors with adenocarcinoma, NEPC, mixed or aberrant phenotypes [95,99] with a prevalence of NEPC after prolonged castration [99]. These findings suggest that primary PC with MYCN amplification may have inherent plasticity and the potential to develop into NEPC after ADT [99]. N-Myc overexpression in conjunction with RB1 and PTEN loss leads to aggressive, NE-like tumors. The lineage switch is accompanied by alterations in chromatin accessibility and redirection of N-Myc binding to NE-associated genes [100], further supporting N-Myc’s role in conferring lineage plasticity properties to the tumor cells.

In PC, MYCN amplification is nearly always concurrent with amplification of Aurora kinase A (AURKA) [37], a serine/threonine kinase that regulates mitotic division and has oncogenic functions in various malignancies [101]. AURKA amplification has been detected in prostate intraepithelial neoplasia (PIN) lesions and may represent an early oncogenic event in PC [102]. AURKA seems to also mediate resistance in CRPC, and its expression is analogous to disease progression [103]. Other mechanisms of AURKA overexpression, besides amplification, may be involved in PC oncogenesis [37,38]. N-Myc and AURKA stabilize each other by forming a complex [37,95,104] and cooperate in the induction of NE features in PC [37]. Combined AURKA and MYCN amplification has also been identified in NE carcinomas of other sites [105]. As expected, both MYCN and AURKA amplification in primary PC predict the transformation to t-NEPC [37,105] (Table 2), independent of other factors such as tumor stage, PSA levels, or GS. MYCN and AURKA amplifications arise early and may even be present in GS-6 tumors that could otherwise be dealt with active surveillance [105]. 

The use of MYCN and AURKA as biomarkers, apart from risk stratification purposes, could have therapeutic implications [105] (Table 2). Although clinical trials of AURKA inhibitors either in unselected CRPC or in AVPC patients did not show significant clinical efficacy [93,106], rare responders exhibited AURKA and MYCN overexpression [93]. N-Myc defies direct therapeutic targeting because, firstly, it is an intrinsically disordered protein [107] and secondly, its structure lacks “druggable” pockets [108]. However, some AURKA inhibitors that alter AURKA conformation destabilize N-Myc too [108,109]. The sensitivity of MYCN amplification to poly (ADP-ribose) polymerase 1 (PARP1) inhibition in neuroblastoma [110] and the identification of the MYCN-PARP1/2-DNA damage repair (DDR) pathway as a driver of transition to NEPC provide the rationale for combining AURKA inhibitors with PARP1 inhibitors. More specifically, N-Myc has been identified as a direct transcriptional activator of PARP1 and PARP2, which in turn regulate the expression of DDR-related genes [111]. Compounds that indirectly target MYCN by disrupting its heretodimerization with myc-associated factor X (MAX), for example, have been developed [112,113]. Recently, a dual inhibitor of N-Myc and AURKA effectively constrained cellular growth in the cell lines of PC and NEPC [114].

Diagnostically, MYCN and AURKA amplifications, although highly specific, are detected in only 20% and 25% of AVPC patients, respectively (Table 3). Another limitation is the lack of established criteria for defining AURKA overexpression by the IHC [24]. The use of IHC as a surrogate of MYCN amplification remains also widely unexplored, besides some evidence showing nearly complete agreement between N-Myc IHC and florescence in situ hybridization (FISH) [115].

## 6. DNA Damage Repair (DDR) Pathway

A significant percentage (20–30%) of mCRPC harbor germline or somatic DDR alterations, associated with adverse prognosis [116,117,118], among which breast cancer 2 (BRCA2) defects are the most prevalent [74,119]. BRCA1/2 defects are responsible for defective homologous recombination (HR) of DNA double-strand breaks (DSBs) [120]. A synthetic lethality approach to PARP inhibition in HR-defective PCs has been proven effective, and PARP inhibitors Olaparib and Rucaparib have been approved by the FDA in this context [121,122,123,124] (Table 2). Furthermore, the presence of HR and especially BRCA2 defects is associated with responses to platinum-based chemotherapy [23,125,126,127,128] (Table 2). 

Among DDR defects, HR alterations are the most common and therapeutically relevant in PC. However, single-nucleotide pleiomorphisms (SNPs) in genes involved in alternative DDR pathways, including mismatch repair (MMR), nonhomologous end joining (NHEJ), base excision repair (BER), and nucleotide excision repair (NER), have been associated with an increased risk of PC, yet with an uncertain causative role [129]. 

Although alterations in MMR proteins are uncommon in PC [74,130], they have been correlated with disease progression [130] and a hypermutated phenotype of advanced PC [131,132]. The detection of MMR germline mutations in a minority of these cases suggests that a subset of MMR-deficient PCs is associated with Lynch syndrome [130,133]. MMR defects and microsatellite instability (MSI) predict response to Programmed cell death 1 (PD-1) blockage [130] (Table 2) in contrast to modest response rates in unselected mCRPC patients [134].

Defects in the NER pathway have been reported to increase PC risk [135]. However, they are not enriched in metastases compared to primary PC [129]. On the contrary, NHEJ, defects tend to increase in metastatic disease [129]. DNA-dependent protein kinase (DNA-PK) mediates NHEJ and its overexpression has emerged as an independent poor prognostic factor, driving PC progression and metastasis [136]. DNA-PK inhibition proved useful in re-sensitizing DU-145DxR PC cells to taxanes [137] and has been tested in combination therapy for mCRPC in a clinical trial (NCT02833883). However, given the additional role of DNA-PK in facilitating AR-mediated transcription [136,138], the prognostic and therapeutic value of DNA-PK in AR-independent PC remains uncertain.

Another DDR effector possibly carrying prognostic information is 8-oxoguanine DNA glycosylase (OGG1), responsible for initiating the excision of 8-oxoguanine, a product of oxidative DNA damage, as part of the BER response [139]. The observed polymorphisms of the OGG1 gene in PC cell lines may compromise the protein’s function, leading to impaired incision of 8-oxoguanine [140]. The Cys allele of the OGG1 gene has been associated with an increased risk of PC and poorly-differentiated, metastatic tumors [141]. 

Screening for DDR defects in AVPC could be predictive of sensitivity to novel therapies. The efficacy of PARP inhibition and anti-PD-1 therapy for AVPC patients is currently being investigated (NCT04592237). DDR alterations were reported to be mutually exclusive with the histologic diagnosis of t-SCPC [40]. Conflicting results show MutS homolog 2 (MSH2) loss in 5% of NEPC [142] and approximately 30% incidence of BRCA2 alterations in AVPC-c [93] (Table 3), in line with the association of BRCA2 mutations with low PSA levels in metastatic PC [143]. DDR defects are biologically related to the AVPC phenotype. The frequently observed BRCA2 and RB1 co-deletion [73,144] drives aggressive behavior and the acquisition of ADT resistance and an epithelial-to-mesenchymal transition (EMT)-like state in PC [144]. Regardless of the presence of DDR alterations, AVPC seems to have inherent genomic instability [24]. p53 and PTEN defects accelerate the cell cycle and do not allow enough time for DDR [23,145]. AURKA can also impair DDR [146]. It is yet unclear if predictive biomarkers could further specify which patients in the AVPC group would benefit from therapies associated with DDR deficiency. 

## 7. Gene Expression Profiles, Epigenetic Regulators and Transcription Factors

NEPC displays a downregulation of AR-mediated and epithelial differentiation genes and overexpresses the NE lineage, EMT, cell cycle, and E2F target genes [37,38,40,41,43,62,147,148,149,150] (Figure 2). Paternally expressed gene 10 (PEG10), a gene of placental development with inherent oncogenic properties, is reactivated in NEPC and represents an attractive biomarker for early NEPC detection (Table 2) and a potential therapeutic target due to the absence of expression in normal adult tissues [151]. The upregulation of mitotic and proneural genes has been validated in AVPC [24]. Gene expression classifiers predictive of NEPC have been developed by different research groups [40,41,43,148,149,150] (Table 2). The advantage of these classifiers as biomarkers stems from their implementation on the limited sampled metastatic tissue that is the subject of extensive IHC studies. The 70-gene NEPC classifier developed by Beltran et al. accurately recognizes NEPC. However, in 20% of cases, elevated scores corresponded to adenocarcinomas, were hypothesized to represent tumors in transition to NEPC, were predisposed for NEPC transformation [43] (Table 2), or simply highlighted the fact that morphology alone is not fully predictive of the behavior of the neoplasm.

The differential gene expression of NEPC is epigenically regulated. Several studies have highlighted diverse epigenetic programs between AVPC and non-AVPC CRPCs. Similar to gene expression, an absolute correlation between AVPC-related epigenome and NEPC morphology has not been seen. For example, the DNA methylation pattern of NEPC can be shared by cases with a morphological diagnosis of PCA but with AVPC clinical features [43] (Table 2), suggesting that epigenetic features may help distinguish AVPC irrespective of morphology. 

It has been postulated that epigenetic alterations are responsible for the lineage plasticity of the AVPC phenotype. RB1 and TP53 loss, frequent molecular events in AVPC, enable epigenetic reprogramming by SOX2 and Enhancer of zeste homolog 2 (EZH2) and the adoption of a stem cell-like program, permissive of lineage switching [62]. EZH2 is a histone methyltransferase member of the Polycomb Repressor Complex 2 (PRC2), which suppresses the transcription of developmental regulators and maintains stemness [152,153] dictating poor outcomes in PC [154]. EZH2 has been associated with AR independence and has been proposed as a prognostic biomarker in PC [155]. More recently, EZH2 upregulation has been particularly associated with t-NEPC [37,43,156,157] and has been found essential for the transformation of PCAs into NEPCs [156] (Table 2). EZH2 synergizes with N-Myc in the induction of the NEPC transcriptional program [95] and is a central downstream mediator of multiple pathogenetic events in NEPC [156]. The oncogenic role of EZH2 is indicative of the benefit of EZH2 inhibitors in combinational therapies [156,158,159,160]. EZH2 expression could be predictive of response to such therapies [158] (Table 2). In pre-clinical studies, EZH2 inhibitors demonstrate an anti-tumor effect on PC and, preferentially, NEPC cell lines [43,161] and re-sensitize NEPC cells to ADT [62]. Clinical trials are currently being conducted in order to investigate the efficacy of EZH2 inhibitors alone or in combination with abiraterone, enzalutamide, or the PARP inhibitor Talazoparib in CRPC patients (NCT04179864, NCT03480646, NCT04846478, and NCT03460977). In addition to EZH2, Clermont et al. identified Chromobox 2 (CBX2), a member of the PRC1, as a frequently overexpressed epigenetic regulator in NEPC and developed a “neuroendocrine-associated repression signature” (NEARS). This signature showed an enrichment for polycomb group (PcG)-silenced genes and was associated with patients’ prognosis [162]. The significance of PcG proteins in AVPC has also been highlighted by the role of PRC1 in forming an immunosuppressive and angiogenetic microenvironment favoring metastases in DNPC [163].

Other epigenetic regulators besides PcG proteins have been implicated in AVPC pathogenesis and could be potential biomarkers and therapeutic targets. DNA methyltransferases (DNMTs) are upregulated in NEPC [43,162,164] and synergize with PcG proteins [162,165] in the induction of a stem cell-like chromatin state [164], which renders tumor suppressor genes vulnerable to silencing [166]. Similar to EZH2 inhibitors, DNA hypomethylating agents restore AR dependence in PC cells [167]. DEK, a DNA topology modulator, is overexpressed in NEPC [40,43,168] and regulates neural genes and genes associated with proliferative and migratory potential. DEK expression in a small percentage of hormone-naïve PCs is an independent poor prognostic factor and could therefore be informative of a propensity towards NEPC transformation [168] (Table 2). Heterochromatin protein 1a (HP1a) expression is an early and persistent event in NEPC development that drives the NE phenotype after castration (Table 2), via repression of AR and (RE)-1 silencing TF (REST). HP1a is part of a heterochromatin gene signature that distinguishes NEPC from PCA [169]. 

Noncoding RNAs (ncRNAs) are regulatory molecules of gene expression with an emerging role in determining tumor phenotypes, including NEPC transformation [170]. Several microRNAs (miRNAs), short ncRNAs with a post-transcriptional regulatory function, have been identified as drivers of NE differentiation and candidate diagnostic biomarkers of NEPC [18,171,172,173]. The observation of the altered miRNA expression profile in NEPC [174,175] led to the development of a miRNA-based classifier that distinguishes CRPC-NEPC from CRPC-Adeno (Table 2). The practical advantage of this classifier compared to gene-expression and mRNA-based classifiers is the stability of miRNAs in formalin-fixed tissues [175]. AR-negative CRPC/ NEPC transcriptome is also characterized by a distinct compilation of long ncRNAs (lncRNAs) with diagnostic and prognostic value [176,177] (Table 2). These lncRNAs embody an additional epigenetic mechanism facilitating lineage plasticity and NEPC induction, partly via interacting with PRC2 [165,176,177].

Transcription factors’ aberrations have also been shown in AVPCs. Loss of REST repression on neural lineage genes has been reported in NEPC and mixed PCA-NEPC [147]. The defect in REST function is, at least in part, mediated by alternative splicing of the respective mRNA by serine/arginine repetitive matrix 4 (SRRM4) [45,178,179]. The alternative splicing fingerprint of NEPC includes a number of SRRM4-regulated genes. SRRM4 emerges as a master regulator that, in AR-depleted conditions, orchestrates the transcriptional and epigenetic modifications needed for transformation into NEPC [178]. REST downregulation induces NE marker expression [178,179,180] but results in amphicrine tumors with AR co-expression and probably sustained sensitivity to ADT. Additional expression of proneural transcription factors (TFs) is required for definite conversion to NEPC [45,179]. In AVPC PDX models, REST presence did not prevent the expression of proneural TFs; however, the possibility of inactive REST splice variants could not be excluded [24].

Proneural TFs reported to mediate transition to NEPC include SOX2, Achaete-scute homolog 1 (ASCL1), POU domain, class 3, transcription factor 2 (POU3F2/BRN2), POU3F4/BRN4, Forkhead box protein A1 (FOXA1), INSM1, and Neurogenic differentiation 1 (NEUROD1) [40,55,100,181,182,183,184,185,186,187] (Table 2). ASCL1 upregulation is an early occurrence following enzalutamide treatment [182] (Figure 2) and can predict aggressive disease [188]. ASCL1 remodels the chromatin architecture and directs PC identity towards a neuronal and stem cell fate, acting in concert with EZH2 to promote lineage plasticity [182]. BRN2, encoded by the AR-repressed gene POU3F2, is required for terminal and SOX2-mediated NE differentiation. BRN2 overexpression is observed not only in NEPC but also in PCA with low serum PSA and probably signifies and is predictive of the transition to an AR-independent state after ADT [183]. BRN4 has been recently identified as an inducible TF upon ADT that cooperates with BRN2 in the initiation of the NEPC program. In fact, BRN2 and BRN4, excreted in the form of extracellular vesicles, could serve as predictive serum biomarkers of NEPC [184]. FOXA1 is a TF that mediates prostate development but remains essential in NEPC as it relocates to NE regulatory elements [185]. INSM1 emerges as a highly specific marker of NEPC [186,189,190] and its expression coincides with Yes-associated protein (YAP) silencing; thus, these two markers could be complementary in the prediction of NEPC [186]. Interestingly, NEUROD1 and ASCL1 expression characterize distinct coexisting subpopulations in NEPC [187]. Wang et al. described a time-dependent expression of TFs and NE genes during the transition to NEPC. ASCL1, as a pioneer factor, governs the initial oncogenic phase but is lost during the late phase of NE differentiation [191]. Expression of TFs SOX2 and POU3F2 and terminal NE markers (chromogranin A and B, enolase 2) are late events in the transdifferentiation process [151,191] (Figure 2). The spatial and temporal heterogeneity of proneural TF expression probably explains why none of those TFs were consistently expressed in all AVPC samples [24] (Table 2). 

## 8. Conclusions

A downside of potent AR inhibition in PC is that it drives, via selective pressure mechanisms, the emergence of AR-indifferent tumors. Unfortunately, morphology cannot definitively distinguish conventional PCA from cases that will follow an unusually aggressive clinical course and could benefit from alternative treatments. Therefore, the need for biomarkers capable of early prediction of AVPC is imperative. 

The concurrence of tumor suppressors’ alterations is a well-characterized but not uniformly present feature of AVPC. A number of additional molecular events, including proto-oncogenes activation and epigenetic modifications, seem to converge into the acquisition of the AVPC phenotype. This is further complicated by the fact that the transition into AVPC is a dynamic process with a time-dependent recruitment of contributing factors. The task of identifying sensitive and specific biomarkers for AVPC detection is thus quite challenging. 

Currently, the implementation of prognostic biomarkers into routine PC diagnostics is lacking. It seems reasonable to suggest a step-by-step approach for their introduction into clinical practice. IHC for AR, with or without IHC for NE markers, could be used as an initial screening method in heavily treated high-grade PCA in order to identify at least a part of AR-indifferent tumors. Combined IHC studies for RB1, PTEN, and TP53 could also pinpoint tumors with the molecular signature of AVPC. As a second step, molecular testing for those tumor suppressors, as well as MYCN and/or AURKA amplification, might prove useful for diagnostic and therapeutic purposes. Similarly, testing for DDR defects may widen the therapeutic choices for treating AVPC patients. However, the benefit of such an approach remains only speculative without the necessary validation from clinical studies. Our increasing awareness and understanding about AVPC are capable of revolutionizing the current standard of care, and, in this setting, biomarkers could be valuable tools in risk stratification and clinical decision-making in the management of CRPC.

## Figures and Tables

**Figure 1 cancers-16-00805-f001:**
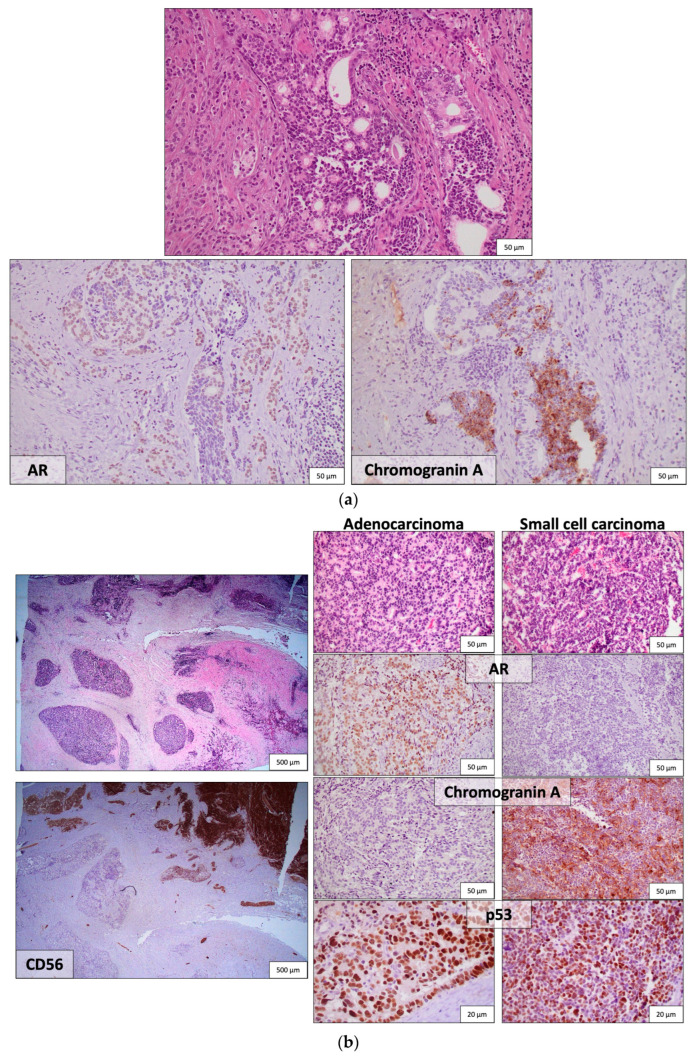
Representative histologic images from two AVPC cases (**a**,**b**) with mixed adenocarcinoma and small cell carcinoma. (**a**) The two components are admixed. AR is expressed in the adenocarcinoma component, and chromogranin A is expressed in the small cell carcinoma; (**b**) The two components are separate within the tumor, as demonstrated by the intense expression of the neuroendocrine marker CD56 only in the small cell carcinoma. Their distinctive morphology and immunohistochemical profiles are depicted in the right panel, yet both show intense p53 expression, consistent with TP53 mutation. [AR = Androgen receptor, AVPC = Aggressive variant prostate cancer, TP53 = Tumor protein 53].

**Figure 2 cancers-16-00805-f002:**
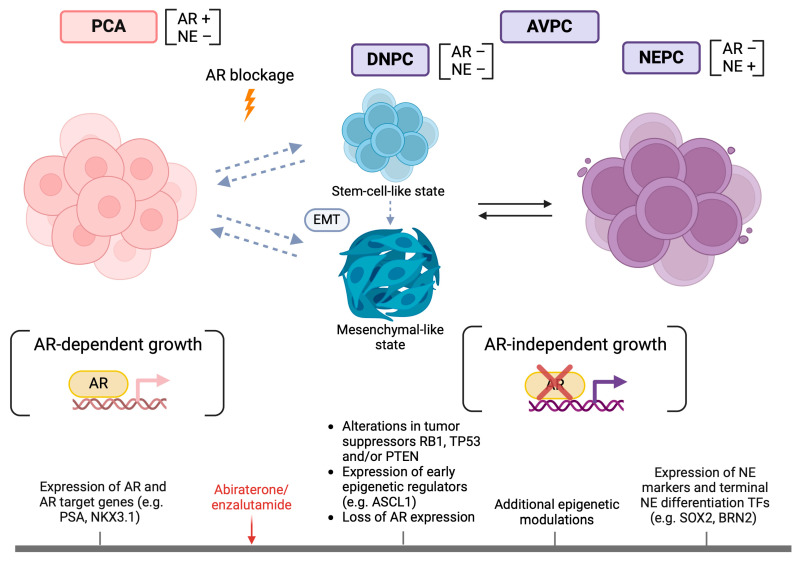
Schematic representation of the transition of PCA to NEPC with the accompanying molecular and phenotypic changes. During transition to NEPC, PCA under the effect of AR blockage (abiraterone/enzalutamide) switches from an AR-dependent to an AR-independent state. An intermediate phase of double negativity for AR and NE markers (DNPC) indicates transient reversal to an undifferentiated, stem-cell-like state. The acquisition of a mesenchymal program marks EMT. The state of DNPC is flexible and can probably be reversed. On the molecular level, combined alterations of tumor suppressors and the recruitment of pioneer epigenetic regulators initiate AVPC emergence. Additional epigenetic events potentiate NE differentiation with the expression of late proneural TFs and NE markers. The reliance of NEPC emergence on epigenetic modifications indicates a probably reversable process. [AR = Androgen receptor, ASCL1 = Achaete-scute homolog 1, AVPC = Aggressive variant prostate cancer, BRN2 (POU3F2) = POU domain class 3 transcription factor 2, DNPC = Double negative prostate cancer, EMT = Epithelial-to-mesenchymal transition, NE = Neuroendocrine (markers), NEPC = Neuroendocrine prostate cancer, PCA = Prostate adenocarcinoma, PTEN = Phosphatase and tensin homolog, SOX2 = SRY-Box transcription factor 2, RB1 = Retinoblastoma protein 1, TP53 = Tumor protein 53].

**Table 1 cancers-16-00805-t001:** Clinicopathologic AVPC criteria compared to features of conventional PCA.

AVPC	Conventional PCA
(1) SCPC histology	Adenocarcinoma histology
(2) Exclusively visceral metastatic spread	Mostly bone metastases
(3) Predominantly lytic bone metastases	Predominantly osteoblastic bone metastases
(4) Bulky (≥5 cm) lymph node mass or bulky (≥5 cm) mass in prostate/pelvis with GS ≥8	Variably bulky disease and variable GS
(5) Low PSA (≤10 ng/mL) at first presentation (before ADT) or at symptomatic progression during ADT despite high volume (≥20) bone metastases	Variable PSA levels at presentation and usually higher PSA levels (PSA > 10 ng/mL) at disease progression
(6) Positive IHC for NE markers (chromogranin A or synaptophysin) or abnormally elevated serum NE markers [chromogranin A or gastrin-releasing peptide) at initial presentation or progression together with non-otherwise explained serum LDH and/or CEA ≥ 2× upper normal value and/or malignant hypercalcemia	Focal/no immunohistochemical expression of NE markers and absence of abnormally elevated serum NE markers and non-otherwise explained elevation of serum LDH, CEA or calcium
(7) Short interval period (≤6 months) between ADT initiation and AR-independent progression	Longer interval time (>6 months) between ADT initiation and progression

Except for patients with a histologic diagnosis of SCPC, all others are required to have undergone ADT and have progressed or had an unsatisfactory response during treatment. [ADT = Androgen deprivation therapy, AR = Androgen receptor, AVPC = Aggressive variant prostate cancer, GS = Gleason score, IHC = Immunohistochemistry, NE = Neuroendocrine, PCA = Prostate adenocarcinoma, PSA = Prostate specific antigen, SCPC = Small cell prostate cancer].

**Table 2 cancers-16-00805-t002:** Candidate tissue-based AVPC biomarkers with their respective diagnostic, prognostic, or predictive value.

Candidate Tissue-Based AVPC Biomarkers	Value
NE markers	Limited value in the absence of NE morphology
AR and AR-regulated genes (PSA, TMRSS2, NKX3.1)	Loss of expression supports AVPC diagnosis but retained expression and transcriptional activity are not preclusive of AVPC
Combined NE and AR expression	AR−/NE− and AR−/NE+ phenotypes are consistent with AVPC
RB1	RB1 alterations associated with poor prognosis Predictive of transition to AVPC
PTEN	Limited value in the absence of concurrent RB1 or TP53 alterations
TP53	TP53 alterations associated with poor prognosis, but are not specific for AVPC
Combined alterations in RB1, PTEN and/or TP53 (≥2/3)	Highly suggestive of AVPC (molecular signature of AVPC)
MYCN	Amplification predictive of AVPC/poor response to chemotherapy (docetaxel)/response to alternative treatments (i.e., AURKA inhibitors)
AURKA	Amplification predictive of AVPC/response to alternative treatments (i.e., AURKA inhibitors)
BRCA2	BRCA2 defects predictive of response to platinum-based chemotherapy/response to PARP inhibitors
MMR proteins	Loss of MMR expression predictive of response to anti-PD-1 therapy
PEG10	Predictive of NEPC
NEPC gene expression classifiers	Diagnostic of NEPC and possibly of AVPC in general
DNA methylation profile	Probable diagnostic value of AVPC, irrespective of morphology
EZH2	Predictive of transition to NEPC/response to EZH2 inhibitors
DEK	Associated with poor prognosis Predictive of transition to NEPC
HP1a	Predictive of transition to NEPC
ncRNA classifiers	Diagnostic of NEPC and possibly of AVPC in general
Proneural TFs (SOX2, ASCL1, BRN2, BRN4, FOXA1, INSM1, NEUROD1)	Predictive of NEPC (taking into account their differential pattern of expression during transition to NEPC)

AR = Androgen receptor, ASCL1 = Achaete-scute homolog 1, AURKA = Aurora kinase A, AVPC = Aggressive variant prostate cancer, BRCA2 = Breast cancer 2, BRN2 (POU3F2) = POU domain class 3 transcription factor 2, BRN4 (POU3F4) = POU domain class 3 transcription factor 4, EZH2 = Enhancer of zeste homolog 2, FOXA1 = Forkhead box protein A1 (FOXA1), HP1a = Heterochromatin protein 1a, INSM1 = Insulinoma-associated protein 1, MMR = Mismatch repair, MYCN = MYCN Proto-Oncogene, BHLH Transcription Factor, ncRNA = noncoding RNA, NE = Neuroendocrine (markers), NEPC = Neuroendocrine prostate cancer, NEUROD1 = Neurogenic differentiation 1 NKX3.1 = NK3 homeobox 1, PARP = Poly (ADP-ribose) polymerase, PEG10 = Paternally expressed gene 10, PSA = Prostate-specific antigen, PTEN = Phosphatase and tensin homolog, RB1 = Retinoblastoma protein 1, SOX2 = SRY-Box transcription factor 2, TFs = Transcription factors, TMRSS2 = Transmembrane serin protease 2, TP53 = Tumor protein 53.

**Table 3 cancers-16-00805-t003:** Candidate tissue-based biomarkers for AVPC detection with their respective sensitivity and specificity values.

**Candidate Tissue-Based AVPC Biomarkers**	**Method**	**Evaluation**	**Sensitivity**	**Specificity**
Chromogranin and/or synaptophysin	IHC	Any extent of positive staining	57% [20]	0–90% [22] ^1^
AR	IHC	Reduced (<10%) or weak (1+) staining	36% [24]	87% [40] ^2^
	Copy number analysis	Absence of copy number gain	80% [24]	30–50% [92] ^3^
RB1	IHC	Reduced (<10%) staining	61% [24]	26–93% [49,50] ^4^
	Copy number analysis	Copy number loss	54% [24]	72% [76] ^5^
p53	IHC	≥10% staining	41% [24]	≈60% ^6^
PTEN	Copy number analysis	Copy number loss	48% [24]	23% [89] ^7^
RB1, TP53 and/or PTEN (≥2/3)	DNA sequencing ^8^	Combined alterations	48% [24]	74% [24] ^9^
MYCN	Copy number analysis	Copy number gain	20% [24]	96% [37] ^10^
AURKA	Copy number analysis	Copy number gain	25% [24]	95% [37] ^11^
BRCA2	DNA sequencing	Mutation or deletion	29% [93]	87% [74] ^12^

^1,10,11^ compared to unselected PCA ^2,12^ compared to unselected mCRPC ^4^ compared to unselected CRPC (26% specificity) [50] or PCA in general (93% specificity) [49] ^3,5,7,9^ compared to unselected CRPC ^6^ estimated by considering p53 IHC as a surrogate for TP53 mutations in unselected CRPC [74,76] ^8^ IHC with a standardized evaluation approach is acceptable [47] [AR = Androgen receptor, AURKA = Aurora kinase A, BRCA2 = Breast cancer 2, IHC = Immunohistochemistry, MYCN = MYCN Proto-Oncogene, BHLH Transcription Factor, PTEN = Phosphatase and tensin homolog, RB1 = Retinoblastoma protein 1, TP53 = Tumor protein 53].

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
