# Peer review of "Tissue-Based Diagnostic Biomarkers of Aggressive Variant Prostate Cancer: A Narrative Review"

_cancers, 2024, doi:10.3390/cancers16040805_

Round 1

Reviewer 1 Report

Comments and Suggestions for Authors

This is an interesting paper on tissue biomarkers related to AVPC. The authors provide a nice compilation of data from the recent and classic literature. The presentation of data is correct.

The paper can be improved:

1. With the current title it is unclear what is really the goal of the paper. I would suggest modifying the title to include word or words such as diagnostics, prognostic, predictive or so, then assisting the potential reader to understand that the paper is a clinically oriented.

2. Not all the readers are familiar with AVPC, so I would recommend a table with main clinical/pathological/radiological defining criteria for AVPC, and, if possible, compared to conventional PCA.

3. Perhaps a table with of the biomarkers reported in the manuscript with potential utility in defining AVPC would also be of interest.

4. The conclusion is unclear. The authors should try to provide a better conclusion and if so acknowledging that the current role of these markers is very limited if any. The authors might need to become proactive and therefore they must present their suggestions on how and what of the markers can be used and when in the context of AVPC.

Comments on the Quality of English Language

The quality of english is OK. 

Reviewer 2 Report

Comments and Suggestions for Authors

The main question addressed by the research is prostate cancer one of the most common after colorectal and lung male cancer. PC has been mainly detected after 55 years. It should be pointed out here that the society of highly developed countries is in an aging period. Therefore, the diseases related to humans  50+ are at the centre of interest. The above is elucidated directly from the cost of medical care/treatment. Therefore, in the review article entitled: Tissue-Based Biomarkers in Aggressive Variant Prostate Cancer: A Narrative Review, authors take into consideration the potential markers of aggressive PC variant. The lack of suitable biomarkers has been correctly noticed and connected with difficulties of histopathologic differentiation. Moreover, even though the PC in people's minds is a well prognostic cancer the information about AVPC was less propagated and therefore early diagnosis is difficult, especially in the case of a PSA low level. Due to the above the information presented in the article fills the gap in this area. The article is well written and readable and can become suitable material for young adepts of medical science. Moreover, the references are correctly selected and cited. During the review process, I have found some critical remarks:

-             The lack of suitable diet and physical activity as a reason for PC should be mentioned as well as their connection to obesity

-             The part of DNA damage and repair should be extended by a level of 8-oxodG discussion of the defect in NHEJ, BER, and NER machineries should be discussed

-             The difference in metal ion concentration of metal ions or antioxidants in the normal and aggressive variants of PC should be mentioned.

-             The pictures in Figure 1 should be given in higher resolution to show the tissue differences.

In conclusion, after the answer to my critical remarks, I can recommend this article for publication.
